# Evolutionary Dynamics of H5 Highly Pathogenic Avian Influenza Viruses (Clade 2.3.4.4B) Circulating in Bulgaria in 2019–2021

**DOI:** 10.3390/v13102086

**Published:** 2021-10-16

**Authors:** Bianca Zecchin, Gabriela Goujgoulova, Isabella Monne, Annalisa Salviato, Alessia Schivo, Iskra Slavcheva, Ambra Pastori, Ian H. Brown, Nicola S. Lewis, Calogero Terregino, Alice Fusaro

**Affiliations:** 1EU/OIE/National Reference Laboratory for Avian Influenza and Newcastle Disease, FAO Reference Centre for Animal Influenza and Newcastle Disease, Istituto Zooprofilattico Sperimentale delle Venezie, 35020 Legnaro, Italy; imonne@izsvenezie.it (I.M.); asalviato@izsvenezie.it (A.S.); aschivo@izsvenezie.it (A.S.); apastori@izsvenezie.it (A.P.); cterregino@izsvenezie.it (C.T.); 2National Reference Laboratory of Avian Influenza and Newcastle Disease, National Diagnostic and Research Veterinary Medical Institute, 1231 Sofia, Bulgaria; gvgoujgoulova@abv.bg (G.G.); iskra013@gmail.com (I.S.); 3OIE/FAO International Reference Laboratory for Avian Influenza, Swine Influenza and Newcastle Disease Virus, Animal and Plant Health Agency-Weybridge, Addlestone, Surrey KT15 3NB, UK; Ian.Brown@apha.gov.uk (I.H.B.); nilewis@rvc.ac.uk (N.S.L.); 4Department of Pathobiology and Population Sciences, Royal Veterinary College, Hatfield, Hertfordshire AL9 7TA, UK

**Keywords:** Bulgaria, highly pathogenic Avian Influenza, H5N2-H5N8 subtypes, genetic characterization, reassortments, duck farms

## Abstract

The first detection of a Highly Pathogenic Avian Influenza (HPAI) H5N8 virus in Bulgaria dates back to December 2016. Since then, many outbreaks caused by HPAI H5 viruses from clade 2.3.4.4B have been reported in both domestic and wild birds in different regions of the country. In this study, we characterized the complete genome of sixteen H5 viruses collected in Bulgaria between 2019 and 2021. Phylogenetic analyses revealed a persistent circulation of the H5N8 strain for four consecutive years (December 2016–June 2020) and the emergence in 2020 of a novel reassortant H5N2 subtype, likely in a duck farm. Estimation of the time to the most recent common ancestor indicates that this reassortment event may have occurred between May 2019 and January 2020. At the beginning of 2021, Bulgaria experienced a new virus introduction in the poultry sector, namely a HPAI H5N8 that had been circulating in Europe since October 2020. The periodical identification in domestic birds of H5 viruses related to the 2016 epidemic as well as a reassortant strain might indicate undetected circulation of the virus in resident wild birds or in the poultry sector. To avoid the concealed circulation and evolution of viruses, and the risk of emergence of strains with pandemic potential, the implementation of control measures is of utmost importance, particularly in duck farms where birds display no clinical signs.

## 1. Introduction

In 2016, HPAI H5 viruses of clade 2.3.4.4B descending from the Gs/GD lineage caused one of the biggest epidemics in Europe by the number of countries affected and the species diversity of infected wild birds [1]. Different HPAI H5 subtypes were detected after the occurrence of reassortment events with LPAI viruses circulating in wild birds. Within the European territory, alongside the HPAI H5N8, which represented the main circulating subtype, HPAI H5N2, H5N5 and H5N6 were also identified [1]. The latter was subsequently responsible for a new epidemic wave in north-eastern Europe between December 2017 and January 2019 [2,3].

In December 2019 a new incursion of a HPAI H5N8 subtype was reported in Poland [4]. Since then, the virus has spread to 8 central-eastern European countries (Bulgaria, Czech Republic, Germany, Hungary, Poland, Romania, Slovakia, Ukraine), affecting mainly the poultry population, with more than 300 outbreaks notified in Hungary and Poland [5,6,7,8]. The last outbreak was reported in June 2020.

In October 2020, with the beginning of the migration of wild birds to their wintering areas, a new HPAI H5 incursion occurred in northern Europe. Similar to the 2016 epidemic wave, the virus had first been detected further east and arrived across a wide area causing variable mortality in the affected wild bird populations and rapidly spread to most of the European countries [9,10,11,12].

In Bulgaria, the first detection of a HPAI H5N8 virus dates back to December 2016. Since then, many outbreaks have been reported in domestic and wild birds in different administrative regions [13]. A recent work by Venkatesh et al. (2020) describes the 2017/2018 HPAI H5N8 epidemic in Bulgaria. The study reports the identification of two separate viral introductions, one in the north-eastern region of Dobrich and another in central and eastern Bulgaria, and suggests that the duck sector played a critical role in HPAI H5N8 virus spread and maintenance in the country [9]. Here, we describe the genetic diversity of HPAI H5 viruses of clade 2.3.4.4B in Bulgaria, analyzing sixteen newly sequenced H5 viruses identified in the country between 2019 and 2021 and the emergence of a new H5N2 subtype, likely from a local reassortment event.

## 2. Materials and Methods

### 2.1. Sample Collection

Between March 2019 and February 2021, sixteen outbreaks caused by the HPAI H5N8 and H5N2 subtypes were registered in chicken and duck farms in Plovdiv, Lovech and Pleven regions (Table 1). 

The National Diagnostic and Research Veterinary Medical Institute of Sofia (Bulgaria) sent clinical samples (intestines, tracheal and cloacal swabs collected from H5Nx infected mule ducks, layer hens and domestic ducks) as well as infectious allantoic fluids (harvested after inoculating 9–11 day old embryonated specific pathogen free (SPF) hen’s eggs for virus isolation) to the EU/OIE/National Reference Laboratory for Avian Influenza and Newcastle Disease at the Istituto Zooprofilattico Sperimentale delle Venezie in Legnaro (Italy) for genetic characterization. 

### 2.2. Whole Genome Sequencing

Total RNA was extracted using QIAamp Viral RNA Mini Kit (Qiagen, Hilden, Germany) according to the manufacturer’s instructions. Complete genomes were obtained as previously described using the SuperScript™ III One-Step RT-PCR System with Platinum™ Taq High Fidelity DNA Polymerase (Invitrogen, Carlsbard, CA, USA) [14]. Amplicons were purified using Agencourt AMPure XP (Beckman Coulter Inc., Brea, CA, USA), quantified with Qubit™ DNA HS Assay (Thermo Fisher Scientific, Waltham, MA, USA), and mixed in equimolar proportion. Illumina Nextera XT DNA Sample Preparation Kit (Illumina, San Diego, CA, USA) was used to prepare sequencing libraries. Sequencing was performed on the Illumina MiSeq platform (2x250 bp Paired-End; Illumina, San Diego, CA, USA). The read quality was assessed by using FastQC v0.11.2 (https://www.bioinformatics.babraham.ac.uk/projects/fastqc/, accessed on 4 March 2021) and raw data were filtered by removing reads with more than 100 bases with Q score below seven, reads with more than 10% of undetermined (“N”) bases, and duplicated paired-end reads. Illumina Nextera XT adaptors sequences (Illumina, San Diego, CA, USA) were clipped from reads with scythe v0.991 (https://github.com/vsbuffalo/scythe, accessed on 4 March 2021) and trimmed with sickle v1.33 (https://github.com/najoshi/sickle, accessed on 4 March 2021). Complete genomes were generated through a reference-based approach using BWA v0.7.12 (https://github.com/lh3/bwa, accessed on 4 March 2021) [15] and processing the alignments with Picard-tools v2.1.0 (http://picard.sourceforge.net, accessed on 4 March 2021) and GATK v3.5 (https://github.com/moka-guys/gatk_v3.5, accessed on 4 March 2021) [15,16,17]. LoFreq v2.1.2 (https://github.com/CSB5/lofreq, accessed on 4 March 2021) [18] was used to call Single Nucleotide Polymorphisms (SNPs). The consensus sequences were submitted to the GISAID EpiFlu™ database (http://www.gisaid.org, accessed on 26 April 2021) under the accession numbers EPI1780066-EPI1780081, EPI1807268-EPI1807356 (Table 1).

### 2.3. Phylogenetic and Evolutionary Analyses 

Sequences were aligned in MAFFT v7 [19]. Maximum likelihood phylogenetic trees were generated in IQTREE v1.6 (https://github.com/iqtree/iqtree1, accessed on 2 September 2021) [20,21], performing an ultrafast bootstrap resampling analysis (1000 replications). Phylogenetic trees were visualized in FigTree v1.4.2 (http://tree.bio.ed.ac.uk/software/figtree/, accessed on 2 September 2021).

The evolutionary rate and the time to the most common recent ancestor (tMRCA) for all the eight gene segments were estimated using Bayesian inference. Markov chain Monte Carlo (MCMC) sampling analyses were performed using BEAST v1.10.4 [22]. We employed an uncorrelated lognormal relaxed molecular clock that allows for rate variation across lineages. We used the SRD06 substitution model (HKY85 + Γ4 with two partitions - 1st + 2nd positions vs. 3rd position -, base frequencies and Γ-rate heterogeneity unlinked across all codon positions) [23]. Maximum clade credibility (MCC) trees were summarized using TreeAnnotator v1.10.4 (http://beast.bio.ed.ac.uk/TreeAnnotator/, accessed on 2 September 2021) and visualized in FigTree v1.4.2 (http://tree.bio.ed.ac.uk/software/figtree/, accessed on 2 September 2021).

## 3. Results

In this study, we analyzed the genetic characteristics of the HPAI H5N8 and H5N2 viruses which caused sixteen outbreaks notified in Bulgaria since the beginning of 2019, and specifically in the periods March-April 2019, February-June 2020 and February 2021. The phylogenetic analysis of the hemagglutinin (HA) gene showed that the HPAI H5N8 viruses collected in March-April 2019 (outbreaks 19/1-19/5) and the H5N8 (outbreaks 20/2, 20/5, 20/7) and H5N2 (outbreaks 20/3, 20/6, 20/8) viruses collected in the period February-June 2020 are related to HPAI H5N8 viruses of clade 2.3.4.4B, which have been identified in Bulgaria since the end of 2016. Despite the introduction in December 2019 in east-central Europe of a novel HPAI H5N8, which extensively circulated in Poland, Germany, Hungary, Czech Republic and Slovakia during the first half of 2020, the outbreaks reported in Bulgaria in the same time period were not caused by the strain newly introduced in Europe through wintering migrations of wild birds (Figure 1). 

The first outbreak of 2020 (20/1) occurred in a mule duck fattening farm. The clinical sample and the respective isolate were both confirmed to be co-infected with H5N2 and H5N8 viral subtypes by Real Time PCR (using specific primers both for the N2 and for the N8 subtypes), as well as sequencing. Specifically, next generation sequencing (Illumina) of the clinical sample only determined an H5N2 subtype, which was the prevalent one, while a partial sequence of about 300 nucleotides of the N8 segment was obtained using Sanger sequencing. Differently, next generation sequencing of the isolate clearly showed a mixed H5N2/H5N8 virus population and the complete sequence of both N2 and N8 genes was obtained. 

The analyses of the complete genomes confirm that the 2019–2020 H5N8 belong to the genotype responsible of the 2016 European epidemic (Figure 2), clustering with HPAI H5N8 viruses circulating in Bulgaria in 2017–2018, for all the eight gene segments. Specifically, they can be grouped into three different sub-clusters: the first one includes the HPAI H5N8 viruses collected in 2019 (outbreaks 19/1 to 19/5); the second one is composed of HPAI H5N8 viruses from outbreaks 20/2 (ducks) and 20/5 (mule ducks) from Plovdiv region; the third one contains HPAI H5N8 viruses from outbreaks 20/7 (hens) and 20/9 (hens), respectively from Kurdzhali and Plovdiv regions.

The H5N2 viruses, all identified in Plovdiv region from outbreaks 20/1 (H5N2/N8 co-infected mule ducks), 20/3 (hens), 20/6 (hens) and 20/8 (hens), derive from reassortment events between the HPAI H5N8 and LPAI viruses. Specifically, the HA and M genes (Figure 1 and Appendix A) cluster with HPAI H5N8 viruses circulating in Bulgaria in 2017–2020, showing the highest identity (99.1%–99.3% for the HA gene) with viruses of sub-cluster 3 (Figure 1 and Appendix A); while the N2 gene clusters with a LPAI H4N2 virus identified in ducks in Bulgaria in 2018 (95.5%–95.6% identity) (Appendix A). The internal gene segments of the H5N2 subtype show the highest similarity with different LPAI viruses identified in wild and domestic birds in Eurasia (Appendix A). In particular, the NS gene segment of the Bulgarian H5N2 viruses belongs to allele B, while the H5N8 viruses possess the allele A (Appendix A). Of note, a sample co-infected with H5N8 and H5N2 viral subtypes had already been identified in ducks in Plovdiv region in 2018 (accession numbers EPI1743963-EPI1743964, GISAID EpiFlu database) but its N2 gene is very different from that of the 2020 H5N2 viruses (Appendix A), indicating an independent reassortment event.

To estimate the time of emergence of the HPAI H5N8 and H5N2 viruses in Bulgaria we performed a Bayesian evolutionary analysis of all the eight gene segments, and we estimated the time to the most common recent ancestor (tMRCA). The HPAI H5N8 viruses belonging to the cluster of the 2018–2020 Bulgarian viruses seem to have emerged in the first half of 2017 (Dec. 2016–Aug. 2017) (Figure 3 and Appendix A). Conversely, the reassortant H5N2 viruses collected in Bulgaria in 2020 appear to have been generated between May 2019 and January 2020, with the exception of NS gene segment, for which the limited number of allele B viruses widens the tMRCA estimate to between October 2017 and July 2019 (Figure 3 and Appendix A). The 2020 reassortant H5N2 strain was identified only in duck and chicken farms located in Plovdiv region, suggesting that it likely emerged in this area, which was hardly affected by the H5N8 subtype outbreaks.

In February 2021 a new introduction of HPAI H5N8 viruses of clade 2.3.4.4B was recorded in the northern region of Pleven. The 2021 HPAI H5N8 viruses (outbreaks 21/1-21/3, Table 1) cluster separately from the viruses previously detected in the country and show the highest identity with the HPAI H5N8 viruses, which have been circulating in Europe since October 2020 [12]; in particular the three Bulgarian viruses are almost identical and cluster with HPAI H5N8 viruses identified in Poland, England and Russia (Krasnodar Region) in November 2020 and January 2021 (Figure 1 and Figure 2).

Molecular analysis of the HA protein revealed that the Bulgarian H5N2 viruses obtained from outbreaks 20/1 and 20/3 possess a potential additional glycosylation site at position 70–72 (D70N) (numbering starting from the first methionine), which likely indicates an adaptation of the virus to domestic birds [24]. For the NA protein, potential additional glycosylation sites were identified in A/chicken/Bulgaria/217_20VIR1724-1/2020 (H5N8) from outbreak 20/7 at position 411-413 (E411N). No evidence of known mutations associated with mammalian adaptation were recorded in any of the analysed viruses.

## 4. Discussion

In Europe, the last cases of HPAI H5N8 viruses related to the 2016–2017 epidemic were detected in March and July 2018 in Italy and Russia, respectively [25,26]. After the detection of the HPAI H5N8 in 2016, Bulgaria has continued to identify up to June 2020 H5 viruses related to the 2016–2017 European epidemic in domestic birds. This may indicate an undetected circulation of the virus in resident wild birds, or more likely in poultry sector, in particular in duck farms where the HPAI H5 viruses have shown to be able to circulate causing no or few clinical signs [27,28]. Extremely low biosecurity measures in foie gras farms and persistent circulation of a single strain in this poultry sector have been previously reported in the country [29]. Several studies based on epidemiological and phylogenetic analyses have recognized domestic Anseriformes to play a critical role in HPAI virus persistence, spread and evolution as well as being an important source for virus infection in chickens [30,31,32,33,34].

After France, Bulgaria is one of the main European foie gras producers, with a highly developed commercial poultry sector which includes mule ducks and hybrid ducks raised for foie gras and meat production. Following recovery from the 2016–2017 avian influenza crisis, production underwent a sharp increase in 2018. In 2019, the Bulgarian industry produced about 2600 tonnes of foie gras intended for export mainly to France (71%), Belgium (14%) and Spain (14%) (https://www.itavi.asso.fr/download/10930, accessed on 10 October 2021, Situation de la production et du marché du foie gras en avril 2020, accessed on 20 September 2021). This rapid expansion, not always accompanied by the implementation of risk-appropriate biosecurity measures with only very basic sanitation and hygiene, could represent the ideal environment for avian influenza viruses to be introduced, maintained undetected and reassort, as demonstrated by the identification of two independently originated H5N2 reassortant strains in 2018 and 2020 in one of the regions with the highest duck population density [35].

Of note, the 2020 H5N2 was firstly identified in a farm rearing mule duck, where the H5N8 was also detected. Additional testing would have been useful to elucidate the extent of circulation of the two subtypes in the farm. The subsequent identification of a genetically related H5N2 strain in hens (outbreaks 20/3, 20/6 and 20/8) may indicate a virus transmission between the two poultry sectors, which may have been fostered by the use of vehicles interchangeably employed to transport eggs, fodder and animals from/to different production holdings. However, we cannot exclude that the virus might have circulated undetected also in other poultry sectors (i.e., backyards). The estimation of tMRCA indicated that this subtype may have been generated between the end of 2019 and the beginning of 2020, which clearly indicates that this genotype was detected soon after its emergence, about two years after the introduction in Bulgaria of the H5N8 progenitor strain (first half of 2017 based on tMRCA estimation). However, given the lack of identification of the LPAI progenitor virus, it is difficult to determine whether the reassortment event took place within the mule ducks farm or in another setting (i.e., in another farm or in wild birds). Increasing active surveillance in wild birds may help to shed light on the ecological context which may favor the emergence of reassortant viruses.

The new introduction in February 2021 of a novel H5N8 strain, which has been showing a high propensity to reassort and the ability to acquire mutations associated with mammalian adaptation, is of concern [36]. The implementation of adequate control measures as well as a regular and capillary surveillance in the poultry industry, in particular in the duck farms, also in absence of evident clinical signs is of utmost importance to stop the circulation of the virus in the country. This could help preventing the emergence of novel reassortant viruses, or of variants with unexpected biological properties, which may represent a serious risk for the operators in the poultry sectors, but also for the general public health.

## Figures and Tables

**Figure 1 viruses-13-02086-f001:**
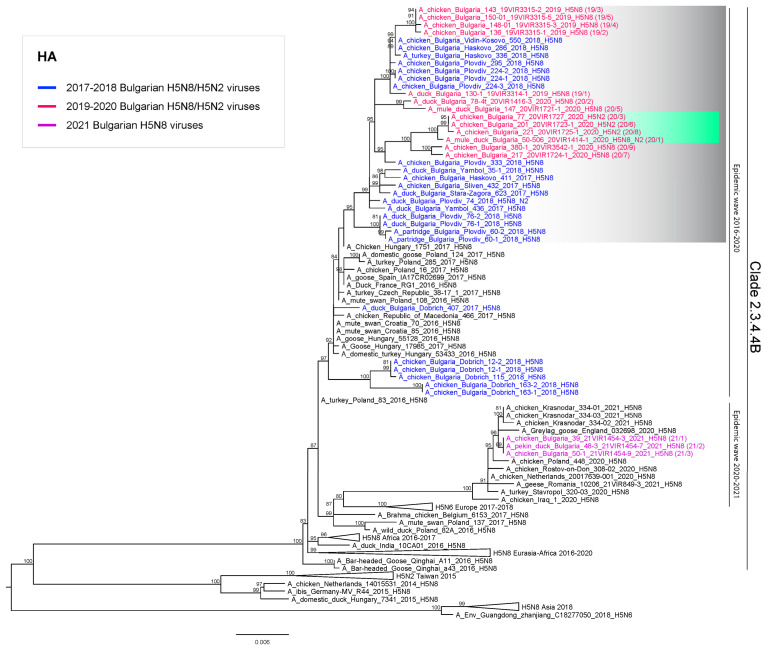
Maximum Likelihood phylogenetic tree of the HA gene (clade 2.3.4.4B). The tree was inferred using IQTREE v1.6.6. The grey box shows the cluster of Bulgarian H5 (N8 and N2) viruses identified in 2019–2020 together with some previously reported 2018 H5N8-H5N2 detections; the light green box shows the cluster of 2020 Bulgarian H5N2 viruses. The viruses are colored according to the years of collection. Ultrafast bootstrap supports higher than 80 are indicated next to the nodes.

**Figure 2 viruses-13-02086-f002:**
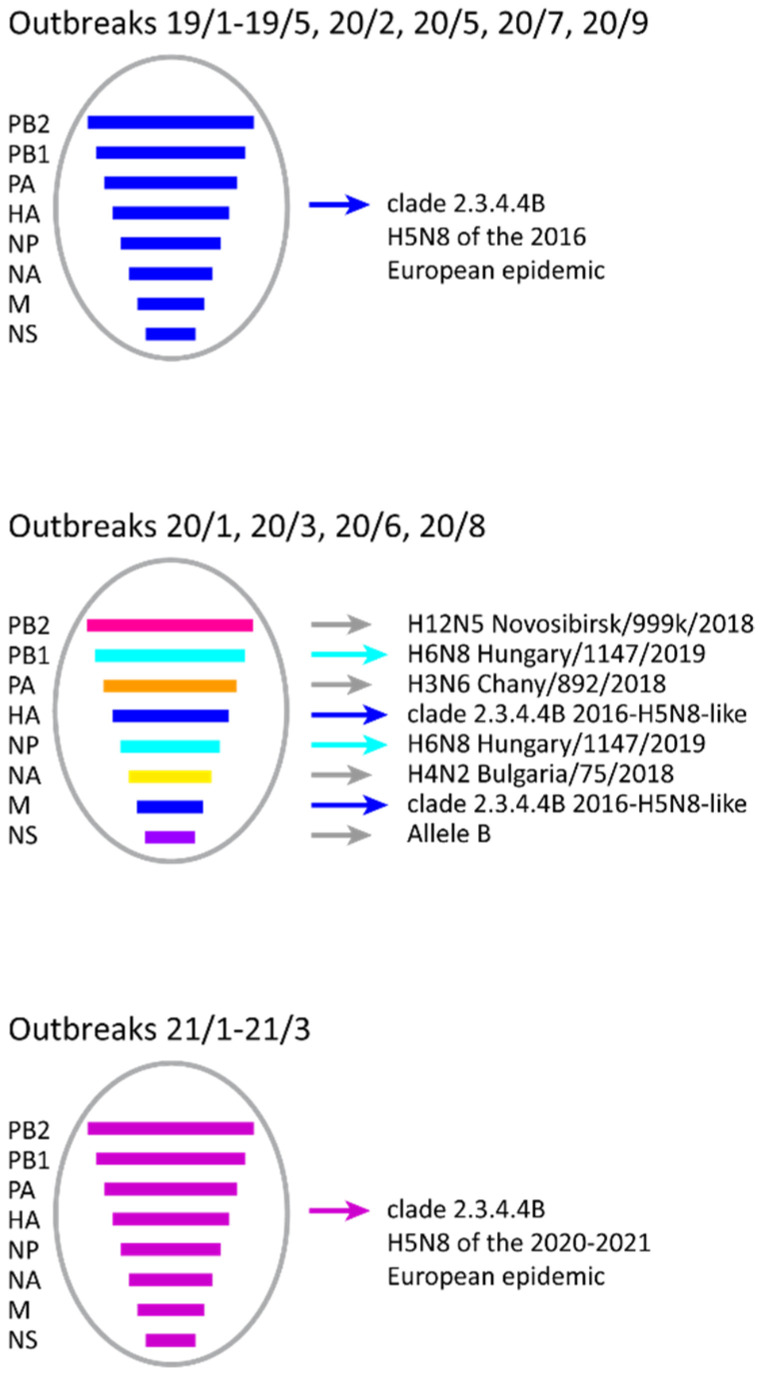
HPAI A(H5) genotypes identified in Bulgaria in 2019–2021. Blue bars: gene segments originating from Eurasian/African HPAI H5N8 viruses of the European epidemic started in 2016; violet bars: gene segments originating from Eurasian/African HPAI H5N8 viruses of the European epidemic of 2020–2021; pink bar: gene segment closely related to the LPAI virus A/mallard/Novosibirsk region/999k/2018; light-blue bars: gene segments closely related to the LPAI virus A/duck/Hungary/1147/2019; orange bar: gene segment closely related to the LPAI virus A/common teal/Chany/892/2018; yellow bar: gene segment closely related to the LPAI virus A/duck/Bulgaria_Plovdiv/75/2018; dark violet bar: gene segment closely related to viruses of which the NS gene belongs to Allele B.

**Figure 3 viruses-13-02086-f003:**
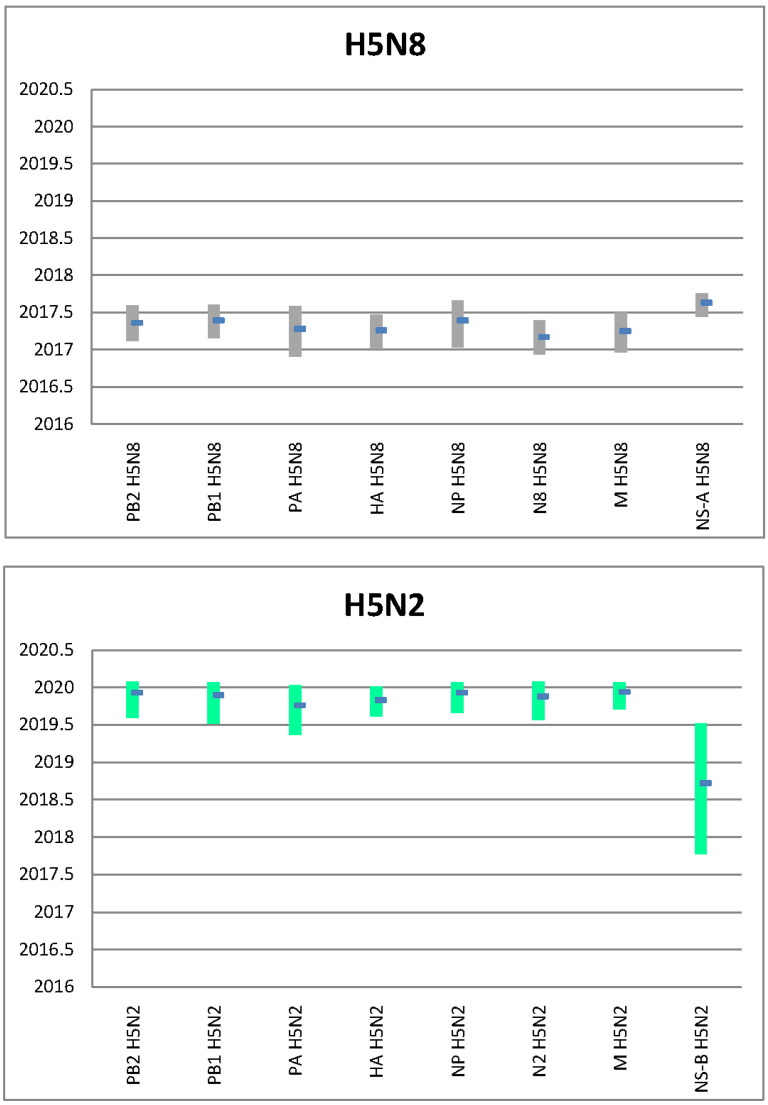
Time to the most common recent ancestor (tMRCA) calculated for each gene segment for the 2019–2020 Bulgarian H5N8 (grey box in Figure 1 and Appendix A) and the H5N2 (light green box in Figure 1 and Appendix A) subgroups.

**Table 1 viruses-13-02086-t001:** Analyzed viruses, epidemiological information and GISAID EpiFlu™ database accession numbers.

Outbreak	Virus	Subtype	Region	District	Collection Date (dd/mm/yyyy)	Species	Production Category	Accession Number (GISAID EpiFlu™)
19/1	A/duck/Bulgaria/130-1_19VIR3314-1/2019	H5N8	Lovech	Yoglav	27/03/2019	Mule duck	Fattening	EPI1807349-EPI1807356
19/2	A/chicken/Bulgaria/136_19VIR3315-1/2019	H5N8	Plovdiv	Krumovo	01/04/2019	Chicken	Backyard	EPI1807268-EPI1807275
19/3	A/chicken/Bulgaria/143_19VIR3315-2/2019	H5N8	Plovdiv	Asenograd	04/04/2019	Chicken	Laying hens	EPI1807276-EPI1807283
19/4	A/chicken/Bulgaria/148-01_19VIR3315-3/2019	H5N8	Plovdiv	Asenograd	05/04/2019	Chicken	Laying hens	EPI1807284-EPI1807291
19/5	A/chicken/Bulgaria/150-01_19VIR3315-5/2019	H5N8	Plovdiv	Asenograd	05/04/2019	Chicken	Laying hens	EPI1807292-EPI1807299
20/1	A/mule_duck/Bulgaria/50-506_20VIR1414-1/2020	H5N2/N8	Plovdiv	Rakovski	11/02/2020	Mule duck	Fattening	EPI1807332-EPI1807339, EPI1807348
20/2	A/duck/Bulgaria/78-4t_20VIR1416-3/2020	H5N8	Plovdiv	Padarsko	21/02/2020	Duck	Foie gras	EPI1807300-EPI1807307
20/3	A/chicken/Bulgaria/77_20VIR1727/2020	H5N2	Plovdiv	Trilistnik	21/02/2020	Chicken	Layer	EPI1780067, EPI1780069, EPI1780071, EPI1780073, EPI1780075, EPI1780077, EPI1780079, EPI1780081
20/5	A/mule_duck/Bulgaria/147_20VIR1721-1/2020	H5N8	Plovdiv	Bolyarino	21/02/2020	Mule duck	Foie gras	EPI1807340-EPI1807347
20/6	A/chicken/Bulgaria/201_20VIR1723-1/2020	H5N2	Plovdiv	Trilistnik	02/03/2020	Chicken	Layer	EPI1807308-EPI1807315
20/7	A/chicken/Bulgaria/217_20VIR1724-1/2020	H5N8	Kurdzhali	Perperek	09/03/2020	Chicken	Laying hens	EPI1780066, EPI1780068, EPI1780070, EPI1780072, EPI1780074, EPI1780076, EPI1780078, EPI1780080
20/8	A/chicken/Bulgaria/221_20VIR1725-1/2020	H5N2	Plovdiv	Manole	11/03/2020	Chicken	Laying hens	EPI1807316-EPI1807323
20/9	A/chicken/Bulgaria/380-1_20VIR3542-1/2020	H5N8	Plovdiv	Asenograd	03/06/2020	Chicken	Laying hens	EPI1807324-EPI1807331
21/1	A/chicken/Bulgaria/39_21VIR1454-3/2021	H5N8	Pleven	Slavyanovo	01/02/2021	Chicken	Laying hens	EPI1858599-EPI1858606
21/2	A/pekin_duck/Bulgaria/48-3_21VIR1454-7/2021	H5N8	Pleven	Slavyanovo	04/02/2021	Pekin duck	Fattening	EPI1858607-EPI1858614
21/3	A/chicken/Bulgaria/50-1_21VIR1454-9/2021	H5N8	Pleven	Slavyanovo	08/02/2021	Chicken	Breeding hens	EPI1858615- EPI1858622

## Data Availability

The consensus sequences of the viruses analysed in this study were submitted to the GISAID EpiFlu™ database under the accession numbers reported in Table 1.

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
