# Peer review of "Evolutionary Dynamics of H5 Highly Pathogenic Avian Influenza Viruses (Clade 2.3.4.4B) Circulating in Bulgaria in 2019–2021"

_viruses, 2021, doi:10.3390/v13102086_

Round 1
Reviewer 1 Report
The manuscript is concise and well-written. I have a few minor suggestions below.
Line 1-3: Consider changing the title to "Duck farms in Bulgaria: a hotspot for the evolution and maintenance of highly pathogenic avian influenza H5 viruses of clade 2.3.4.4B"
Line 81: cryptic spread and maintenance of AIV - Do you mean all AIV or an introduced specific strain?
Line 126. all ESS >200 ?
Line 130 (Table 1): the format of collection date should be consistent (the outbreak 21/1-3 are different from the outbreak 19/1-20/9). Indicate the format (e.g. DD/MM/YYYY) in the table.
Line 136: genetic - phylogenetic
Line 211-214: provide a reference that describing the D70N is related to an adaptation of the virus to domestic birds here.
Discussion: To emphasize the important role of domestic ducks in the evolution, spread, and maintenance of HPAIV, it would be great to add a paragraph regarding the role of domestic ducks in other countries with reference papers. For example, "Many previous studies also have indicated domestic duck as a major source of HPAIV based on phylogenetic and epidemiological analysis......."
Supplemental data: There are only ML phylogenies in the supplemental data. Please provide the Bayesian time-scaled trees used for the estimation of tMRCA as a supplemental file.
Reviewer 2 Report
Duck farms in Bulgaria: a hotspot for the evolution and maintenance of highly pathogenic avian influenza viruses
Authors describe characterization of recent HPAI H5 viruses isolated from different commercial production systems in Bulgaria between 2019-2021 and compare the sequences to other HPAI H5 viruses since 2016.
I do not feel the title accurately represents the information presented in the article. As the authors state in the introduction, this article is more about characterizing the genetic diversity of 2.3.4.4b viruses in the country and doesn’t really contain data showing the duck farms are responsible for that diversity
Check clade designation throughout manuscript – sometimes 2.3.4.4B and sometimes 2.3.4.4b
Check grammar throughout article – there are several misplaced and misused commas which change the meaning or understanding of the sentences
Line 80-81: these are the exact words used in reference 9
Line 91: suggest simplifying the types of samples, e.g. intestines, tracheal and cloacal swabs rather than organs and swabs with specifics in parenthesis. Where were the allantoic fluids collected? Were the viruses propagated in embryonated eggs prior to forwarding? If so, need to include virus isolation in methods. Were the allantoic fluids collected directly from eggs in the field? If so, how were they collected, e.g. age of egg, method for collection? Were the sequences generated direct from sample (e.g. intestine and swab), or were all sequences generated from virus isolates?
Line 98: Capitalize proper name of extraction kit (Mini Kit)
Line 99: add “as previously described” after “obtained” so the reader knows that an amplification protocol was used – this gets lost even though there is a reference.
Line 104: lower case sequencing libraries
Line 108-109: Illumina Nextera XT adaptor sequences were clipped from reads (opposite of how currently written)
Table 1: Check production categories – some “fattering” and some “fattening” are these different?
Phylogenetic trees are very nice and easy to read. Like the highlighting to draw attention to sequences of interest
Lines 151-154: lower case real-time; this sentence raises a lot of questions – what were the Ct values with the N2 and N8 real-time PCR assays? Is it possible the N8 reaction was non-specific or otherwise incorrect? Or was the N8 Ct value consistent with the N2 Ct value – and consistent with a value expected from a virus isolate? When this isolate was sequenced, was there evidence of mixed population in the other gene segments, or were the sequences pure? Would like to see the authors address this inconsistency in the discussion
Line 196: lower case t on tMRCA
Line 222: Introduction states HPAI H5N8 was first detected in 2016, this sentence states 2017. This sentence is difficult to understand as written – suggest rewording to clarify.
Line 241: raring should be rearing
Line 242: See previous comment – if there was only a single isolate from the mule duck farm, would suggest including additional information to support the H5N8 was really present on that farm and not a false positive – a single detection doesn’t equal circulation on the farm. The next sentence states subsequent virus transmission from ducks to hens – that is not presented in the results. Is there an epidemiologic link to support that or is it based on the phylogenetic tree? Suggest including how that conclusion was reached in the results section.
Line 258: avoiding should be prevent
